# Phytochemicals: Potential Therapeutic Modulators of Radiation Induced Signaling Pathways

**DOI:** 10.3390/antiox11010049

**Published:** 2021-12-27

**Authors:** Bimal Prasad Jit, Biswajita Pradhan, Rutumbara Dash, Prajna Paramita Bhuyan, Chhandashree Behera, Rajendra Kumar Behera, Ashok Sharma, Miguel Alcaraz, Mrutyunjay Jena

**Affiliations:** 1School of Life Sciences, Sambalpur University, Jyoti Vihar, Burla 768019, India; bimaljit2019@aiims.edu (B.P.J.); rutulifescience@gmail.com (R.D.); rajendrabehera@suniv.ac.in (R.K.B.); 2Department of Biochemistry, AIIMS, Ansari Nagar, New Delhi 110029, India; sharma@aiims.edu; 3Algal Biotechnology and Molecular Systematic Laboratory, Post Graduate Department of Botany, Berhampur University, Bhanja Bihar, Berhampur 760007, India; pradhan.biswajita2014@gmail.com (B.P.); chhandashreebehera@gmail.com (C.B.); 4Department of Biotechnology, Sangmyung University, Seoul 03016, Korea; 5Department of Botany, Maharaja Sriram Chandra Bhanja Deo University, Baripada 757003, India; prajnabhuyan2017@gmail.com; 6Radiology and Physical Medicine Department, School of Medicine, Campus de Excelencia Internacional de Ámbito Regional (CEIR)-Campus Mare Nostrum (CMN), Universidad de Murcia, 30100 Murcia, Spain

**Keywords:** phytochemicals, radiation effects, therapeutics, signaling pathways, radioprotectors

## Abstract

Ionizing radiation results in extensive damage to biological systems. The massive amount of ionizing radiation from nuclear accidents, radiation therapy (RT), space exploration, and the nuclear battlefield leads to damage to biological systems. Radiation injuries, such as inflammation, fibrosis, and atrophy, are characterized by genomic instability, apoptosis, necrosis, and oncogenic transformation, mediated by the activation or inhibition of specific signaling pathways. Exposure of tumors or normal cells to different doses of ionizing radiation could lead to the generation of free radical species, which can release signal mediators and lead to harmful effects. Although previous FDA-approved agents effectively mitigate radiation-associated toxicities, their use is limited due to their high cellular toxicities. Preclinical and clinical findings reveal that phytochemicals derived from plants that exhibit potent antioxidant activities efficiently target several signaling pathways. This review examined the prospective roles played by some phytochemicals in altering signal pathways associated with radiation response.

## 1. Introduction

In recent years, there has been an increase in public interest within the scientific community regarding the hazardous effects of ionizing radiation. Ionizing radiation (IR) has become an intrinsic component of several industries, including nuclear power, agricultural, and medicinal industries. Different forms of IR, including α particles, β particles, protons, x, and γ-rays, could potentially cause damage to biological systems at the cell, tissue, or body levels [1]. Currently, cancer is seen as one of the leading causes of morbidity and mortality worldwide. Biological systems have inbuilt mechanisms to protect themselves from the harmful effects of low levels of exposure to IR as deployment radiation shielding mechanisms. However, during unavoidable exposures to IR under certain conditions like nuclear accidents and or planned exposures during cancer radiotherapy, the hazardous substances produced by IR are massive and the body’s own defense mechanisms may not be appropriate for protection [2]. In addition to chemotherapy, immunotherapy, hormonal therapy etc., radiotherapy is an important cancer treatment modality. The exposure of tumor cells to IR leads to the induction of free radical species; reactive oxygen species (ROS) and reactive nitrogen species (RNS) which cause DNA damage, lipid peroxidation, and the release of signal mediators (ligands like growth factors, cytokines, and hormones) [3]. Together, these molecular players can activate the prominent paracrine and endocrine signaling pathways, leading to target cell damage and radiation-induced bystander effects [4]. Preclinical and clinical findings in malignant lesions demonstrate that perturbations of signaling pathways in cancer cells play a crucial role in their sensitivity to ionizing radiation [4].

It has been shown that the extent of radiation-induced damage is highly systematic and depend on the type of tissue or organ involved. For instance, the gastrointestinal system is highly susceptible to radiation damage due to its rapidly dividing cells [5]. Radiation injuries, like inflammation, fibrosis, and atrophy, are characterized by genomic instability, apoptosis, necrosis, and oncogenic transformation, which are mediated by the activation or inhibition of specific signaling pathways [6]. Both pro-survival pathways (PI3K/Akt, JAK/STAT, etc.) as well as proapoptotic signaling cascades (Wnt and p53, etc.) play a vital role, subsequently leading to radiation-induced responses [6]. The last decade of research has identified several synthetic and semi-synthetic compounds which have shown promise for use in radiation medicine [7]. Amifostine, the first FDA-approved chemical agent, is a clinically effective chemotherapeutic substance used in normal tissue for radiological insults. Conversely, the inherent toxicity and high cost of synthetic r’ radioprotectors warrants the search for alternative radioprotective agents [8]. Recent reports have shown that naturally occurring compounds, especially phytochemicals, have the potential to modulate signaling pathways [9,10,11,12]. Several epidemiological studies have shown that the intake of some phytochemicals can exert effects on several signalling pathways and reduce the risk associated with radiation damage [13]. Owing to the promising antioxidant properties of phytochemicals, in this review, we emphasize their critical roles in radioprotection and analyze the signaling pathways involved in this process.

## 2. Radiation Damage and Phytochemical Action

IR adversely affects the biological system extensively. Along with surgery and chemotherapy, radiation therapy (RT) is a crucial approach for the treatment and management of cancer [14]. Nuclear accidents, RT, space exploration, and the nuclear battlefield release massive amounts of radiation, leading to the exposure of biological systems. Interaction with IR leads to significant biological consequences characterized by inflammation, radiation-induced fibrosis (RIF), carcinogenesis, and death. Radiation-induced responses are mediated by its direct and indirect effects [15,16]. Directly, the IR can interact with DNA and cause (double) strand breaks on the DNA. An indirect effect is characterized by the generation of free radical species like ROS (^●^OH: hydroxyl radicals, ^●^H, e^−^_(aq),_ H_2_O_2_, H_3_O^+^), as a result of radiolysis of water [17]. These charged species can interact with biological macromolecules, like DNA, RNA, protein, and membrane lipids, leading to cellular dysfunction, damage, and/or death. [18]. Highly reactive oxygen free radicals can induce DNA lesions, contributing to DNA mutation and genome instability. In addition to DNA damage response, mitochondrial and endoplasmic reticulum (ER) stress responses may increase ROS generation [19]. ROS and other IR-induced products inside cells augment the release of diverse cytokines leading to local or systematic effects from cellular to body levels [20].

It has been observed that about 50% of the patients receive RT along with surgery and chemotherapy [15]. Despite the randomness of radiation reactions, the impact of radiation is propagated in a sequence-specific manner, leading to the activation of several signaling targets. It has been shown that multiple signaling pathways, e.g., ATM/TP53, MAPK, and NFkB, could lead to the altered expression of several effector genes, in turn leading to a knock-on effect on cells (Figure 1).

The mechanistic pathways can influence cell cycle regulation, DNA repair, and cell death or apoptosis. Altered activation of signaling pathways is involved in the modulation of pleiotropic cytotoxic and cytoprotective cellular responses, which subsequently lead to the regulation of cell proliferation, senescence, differentiation, and apoptosis [21]. Phytochemicals are known to have a constructive effect on biological systems and play key roles in the treatment and/or management of numerous diseases, including chemo and radiotherapy in cancer. Phytochemicals, owing to their inherent antioxidant abilities, can scavenge free radicals and create a signals in response to electrophile and chemical stress leading to activation or inhibition of several signaling responses. Figure 2 shows different potential intermediate targets of phytochemicals during radiation-induced stress. NF-E2-related factor 2 (Nrf2) signaling is linked to phase II detoxifying enzymes, phase II transporters, anti-oxidative stress proteins, and stress defense molecules. Phytochemicals have the ability to activate Nrf2, which ultimately acts against ROS and RNS or other reactive carcinogenic metabolites [22]. Previous evidence shows that phytochemicals can modulate multiple signaling pathways during stress responses associated with IR [23]. Apigenin (4′,5,7-trihydroxyflavone) is a dietary component that has been found to be radioprotective in lymphocytes [24,25], keratinocytes, and mice models [26]. Under hypoxic conditions, betullinic acid, a triterpene, can function as a radiosensitizer in glioma cells [27]. After exposure to 7–8 Gy of gamma radiation, ascorbic acid at a dosage of 3 g/kg body weight reduces radiation lethality and contributes to mouse survival [28]. Pretreatment with caffeine can significantly inhibit radiation-induced micronuclei formation [29]. Curcumin ([1,7 bis (4-hydroxy-3-methoxyphenyl) 1,6 heptadiene 3,5 dione]), a natural phenol that significantly reduces radiation-induced clastogenicity, decreases ROS production and lipid peroxidation, and inhibits radiation induced genotoxicity [30]. Resveratol (3,4′,5-trihydroxy-transstilbene), a phytoalexin, has potential free radical scavenging activity and is radioprotective against IR [31]. Lycopene, (tetra teepee hydrocatbon), a carotenoid, exhibits free radical scavenging ability and is shown to be effective against radiation-induced chromosomal aberration [32]. In addition, sesamol (3,4-methylenedioxyphenol), a nutritional phenolic compound, is effective against radiation-induced genotoxicity, as well as radiation-induced intestinal and hematopoietic injury due to its antioxidant nature [33]. Furthermore, EGCG (epigallocatechin-3-gallate), a green tea catechin, has been evaluated by several studies for its radiomodulatory ability. [24,34]. Mangiferin, a glycosyl xanthone, successfully reduces radiation-induced mortality, oxidative stress associated with renal injury, and decreases radiation-induced micronucleated binucleate cells [35]. Furthermore, compounds like genistein, vanillin, hesperidin, eugenol, vinblastine, vincristine, orientin, and vicenin, ellagic acid, gallic acid, quercetin, trigonelline, myricetin, naringin, etc. exhibit potential antioxidant, anti-inflammatory, antiproliferative, anti-mutagenic, and radioprotective activities [9,24,36,37]. Therefore, diverse clinical trials and further characterization of these compound are of paramount importance so far as the translational value is concerned.

## 3. Phytochemicals and Their Possible Roles in Radioprotection via Different Signaling Pathways

### 3.1. NFκB Signaling Targetted Pathway

NFκB is a family of dimeric transcription factors that control the transcription of a variety of genes in the promoter regions of specific target genes. The enhancer element of the immunoglobulin kappa light chain is where NFκB binds. NFκB is found in almost every cell type and plays a role in cell proliferation, differentiation, immunological, and inflammatory response [38]. Various studies have shown that exposure to radiation doses ranging from 0.1 to 2 Gy can activate NFκB. The protein binds to a sequence in the immunoglobulin light chain enhancer in B cells, triggering several genes involved in inflammation, cell proliferation, differentiation, and various stress responses [39]. In mammalian cells, the protein encompasses five members of the Rel family, which include Rel A, RelB, c-Rel, NFκB1 (p105/p50), and NFκB2 (p100/p52) [40]. During rest, the protein is inactive in the cytoplasm, and its activity is exclusively dependent on a family of regulatory proteins known as inhibitors of NFκB (IkB). The molecule plays a significant role in resistance to radiation and chemotherapy, being involved in anti-apoptotic activity, cell growth, and clonogenic capacity, as has been observed in numerous human cancer cell lines [41]. Although classical (canonical) and alternative pathways are crucial in activating the NFκB and its translocation to the nucleus, the involvement of unique signaling pathways, type of stimuli, and cell types are the major confounding factors. The classical pathway involves the IKK-β-dependent degradation of IκB-α, IκB-β, and IκB-ε, whereas the alternative pathway involves the IKK-α- dependent activation of p52/Rel B [42]. IR, TNF-α (tumor necrosis factor-α), PMA (phorbol 12-myristate 13-acetate), LPS (lipopolysaccharide), and interleukins are the major stimuli that activate the classical pathway, thus causing IκB degradation [43,44]. Ligands to certain members of the TNF receptor superfamily, such as the B-cell-activating factor of the TNF family, CD40, or lymphotoxin β, can initiate the alternative route, which is fully independent of IKK-β and IKK-γ. The two pathways are the target sites for many natural radioprotectors, culminating in their response against the harmful effects of IR [45].

Several studies have revealed the rescue effect of NF-κB in irradiated cells [46,47]. Once the pathway is activated, it causes the inhibitor subunits IκB-α and IκB-β to be phosphorylated, followed by ubiquitination and degradation, resulting in the release of active NF-κB. NF-κB translocates into the nucleus after activation, where it binds sequence-specifically to the promoter/enhancer region of multiple target genes and transactivates their expression [48]. The presence of one or more NF-κB binding sites in the promoter/enhancer region of TNF-α has been confirmed by multiple laboratories. Studies have demonstrated NF-κB-mediated TNF-α expression by several inducers. IR-induced NF-κB activation modulates anti-apoptotic signaling pathways in conjunction with telomerase activation in a TNF-α mediated manner and thus imparts survival advantage in the bystander cells [49,50]. Drugs targeting NF-κB can inhibit tumor cell proliferation, and are thus considered as anti-tumor agents. The survival effect of NF-κB is exerted by its downstream signaling molecules, e.g., cellular inhibitors of apoptosis (cIAPs), B cell lymphoma family proteins. DIM (3,3′-diindolylmethane) protects against radioactivity by inducing an ATM-driven DDR-like response and the NF-KB signaling pathway [51]. Hesperidin was found to modulate inflammatory targets like NF-κB and thus play an essential role in radioprotection. Although there is a lack of literature regarding the phytochemical action in radiation signaling, accumulating evidence suggests the anticancer activity of many compounds of plant origin that can be used in in vitro and in vivo models for modulating the NF-κB signaling pathway. Parthenolide, a sesquiterpene lactone obtained from fruit and flower of *Tanacetum parthenium*, inhibits NF-kB in MCF-7 cells and thus possesses anticancer activity [52]. The NF-kB also acts as a potential target for phytochemicals like EGCG, pterostilbene, ATRA (All trans- retinoic acid), and curcumin [53]. In LPS-RAW264.7 cells, procyanidins from wild grape seed (WGP) prevent the stimulation of NF-κB and p38 MAPK pathways and thus decrease the oxidative stress-mediated ROS and NO generation [54]. A study by Ren et al. [55] showed the inhibitory effect of resveratrol on NF-KB signaling through the p65 and Ikappa B kinase activity.

### 3.2. Targeting Wnt Signaling Pathway

IR and radiotherapy are potential modulators of the Wnt signaling pathway [56]. The acquisition of radioresistance and the development of invasive phenotypes are both aided by the activation of Wnt/β-catenin. Accumulating evidence suggests that exposure to IR and radiotherapy is intrinsically associated with the RIF, characterized by inflammation, accumulation of high density of unorganized myofibroblasts, retractile fibrosis, and gradual loss of parenchyma cells [57,58,59]. Activation of NF-κB during IR stress plays a significant role in activating the Wnt pathway [60]. Wnt finetunes the cell growth, metabolism, development, and maintenance of stem cells [58]. Inflammation resulting from RT and IR leads to NF-κB generation and the production of TGF-β (transcription growth factor β). TGF-β plays a chief role in the manufacture of fibroblast, which on differentiation can form myofibroblast from the bone marrow progenitor cells [61]. Wnt ligands are activated by inflammation. FZL and LRP5/6 receptors bind Wnt ligands leading to the destruction and inactivation of AXIN/APC/GSK-3β complex. The inhibition of β-catenin phosphorylation prevents it from being degraded by the proteasome. Accumulation of β-catenin in the cytoplasm before translocating to the nucleus to bind TCF/LEF co-transcription factor induces WNT target genes, including c-Myc and cyclin D1. TGF-β1 is activated by inflammation and DNA damage, which activates the Smad pathway. TGF-β1 binds to TGF-β receptor type 2, causing the recruitment of TGF-β receptor type 1. Smad2/3, which binds to Smad4, is phosphorylated by the hetero-tetramer that forms. The Smad complex translocates to the nucleus to activate CTGF and other target genes [62,63].

Dickkopf-1 (DKK) is activated by PPAR γ agonists, which block WNT ligands and prevent β-catenin accumulation by activating GSK-3β [64]. PPAR γ agonists reduce Akt activity while stimulating PTEN, a PI3K inhibitor. Smad7 and PTEN are also stimulated by PPAR γ agonists, which block the Smad pathway [65,66]. Diosmin, a citrus bioflavonoid with antioxidant, anti-inflammatory, and anti-apoptotic characteristics, has been shown to boost PPAR γ expression and inhibit the canonical WNT/β-catenin pathway, which can help to prevent radiation-induced hepatic fibrosis [67]. Through the reduction of NF-κB expression and downregulation of the STAT-3 pathway, PPAR γ activators can prevent irradiation-induced inflammatory processes. It has also been reported that IR induces the WNT/β-catenin signaling pathway via the up-regulation of several downstream genes, such as MMP-2, MMP-9, VEGF, CD 44, and TCF 1, and thus increase the invasive potential of U87 cells [68]. Further evidence provided by Huang et al. (2020) [69] revealed the correlation between IR and promotion of WNT/β-catenin signaling during RT, which significantly augments LIG4 (DNA ligase IV) activity in colorectal cancer cells (CSC) and promotes radioresistance. Several compounds of plant origin have been developed as potential radioprotective agents to modulate Wnt/β-catenin signaling. For instance, fisetin, a flavone found in many plants, such as strawberry, apple, grapes, cucumber, persimmon, onion, *Acacia greggii*, and *Acacia berlandieri*, inhibits Wnt through the expression of β-catenin [70]. Furthermore, resveratrol inhibits wnt signaling in colon cells and significantly decreases the invasiveness of a variety of tumor cells [71,72]. Arthur et al. (2014) [73] revealed that the administration of *Ajuga turkestanica* extract protects from muscle injury by modulating the Wnt and Notch pathways. Sulforaphane, an organosulfur compound, has been observed to downregulate the Wnt/β-catenin self-renewal pathway in breast cancer stem cells and protect skin against UV-induced damage [74].

Extracts of *Ginko biloba* exocarp have been reported to inhibit angiogenesis in Lewis’s lung cancer cells, possibly by acting on the Wnt/β-catenin-VEGF, indicating its possible radiomodulatory effect of the Wnt/β-catenin pathway [75]. Further studies have revealed a radioprotective effect of indigo wood root extracts in alleviating radiation-induced mucositis. Subsequent findings show indirubin, a major phytoconstituent, acts as an agonist of the Wnt/β-catenin pathway [76,77]. Another indirubin derivative, indirubin-3′-oxime, is an agonist candidate for Wnt/β-catenin and plays a significant role in preventing radiation-induced bone injury.

### 3.3. Targeting Nrf2 Signaling Pathway

IR is a multi-faceted stress agent that poses a severe threat to the biological system by producing a diverse amount of free radical species, e.g., ROS and RNS, which induce a variety of responses, such as inflammation, cancer, oxidative stress, and genomic instability [78,79]. To combat the deleterious effect of free radicals, new signaling pathways are induced, which modulates the expression of the antioxidant-responsive elements signaling pathway (induced by genes expression) and acts as the first line of protection against oxidative stress. One of the keys signaling molecules involved in cellular stress response is nuclear transcription factor erythroid 2p45 (NF-E2)-related factor 2 (Nrf2), whose role is of paramount importance in the up-regulation of different antioxidants cytoprotective genes. Nrf2 contains a basic leucine zipper motif that acts as a transcription factor that binds an antioxidant response element (ARE) or an electrophile response element (EpRE). It thus activates phase II/detoxifying and other antioxidant gene expressions by binding to the cis conserved core sequence (5′-A/GTGAC/GNNNGCa/c-3′) situated in the promoter region as well as transactivators and coactivators small Maf-F/G/K and cAMP response element-binding protein (CREB-binding protein or CBP), p300, which regulates the ARE-driven antioxidant gene transcription. Accumulating evidence suggests the potential role of Nrf2 in ARE-mediated gene expression [80]. Glutathione S-transferase (GST), UDP-glucuronosyltransferase (UGT), heme oxygenase-1 (HO-1), NADP(H): quinone oxidoreductase (NQO), glutamate-cysteine ligase (GCL), and gamma-glutamylcysteine synthetase (γGCS) are among the phase II detoxifying and antioxidant enzymes [81]. Under normal/basal conditions, the Nrf2 is tethered in the cytoplasm as a sedentary complex with a cytoskeletal binding protein which acts as a repressor called Kelch-like ECH-associated protein 1 (Keap1), regulating its translocation to the nucleus. Nrf2 can be proteosomally degraded by Cul3–Keap1 ubiquitin E3 ligase complex. Any agent interfering with the interaction of Nrf2 and Keap1 by the covalent alteration or oxidation of cysteine residue of Keap1 protein decreases E3 ligase activity, which subsequently causes the release of Nrf2 which is translocated to the nucleus and binds to ARE and EpRE, thus stimulating the expression of cytoprotective genes [82].

Substantial epidemiological evidence revealed the possible mechanism of action of phytochemicals and their mechanisms of action on Nrf2 signaling. EGCG, a major catechol present in tea, stimulates Nrf2 expression and its translocation to the nucleus. The compound induces HO-1 synthesis in rat neurons and acts as an effective neuroprotective agent [83]. Accumulating evidence suggests that curcumin administration can stimulate the HO-1 pathway by disrupting the Keap1/Nrf2 complex [84]. Curcumin attenuates oxidative stress by modulating Nrf2 signaling. EGCG induces Nrf2 in a PI3K and ERK-dependent manner in human mammary epithelial cells [85], possibly due to its (EGCG) antioxidant activity reported earlier.

Furthermore, EGCG induced HO-1 expression by activating Akt and ERK1/2 in endothelial and MAPK cells (P38) and Akt mediated signaling in B-lymphoblasts [86]. Similar findings showed that feverfew extracts could protect from oxidative DNA damage by making DNA repair in skin cells through the P13K-dependent-Nrf2/ARE pathway [87]. The antioxidant effect of resveratrol in in vivo and in vitro models has been extensively studied. Data show resveratrol stimulates Nrf2 mediated glutathione synthesis in human lung epithelial cells [88]. The compound also stimulates Nrf2 mediated HO-1 synthesis in PC12 cells [89]. Lycopene administration significantly induces antioxidant enzymes, e.g., SOD, GR, and GSH, and decreases the lipid peroxidation marker malondialdehyde (MDA). Subsequent findings showed that the administration of zerumbone, a sesquiterpene derived from zinger, induces Nrf2 signaling and the expression of its target protein HO-1 in mouse epidermal cells [90]. Quercetin activates Nrf2 expression and down-regulates Keap1, thus inducing the Nrf2 mediated ARE pathway in an ERK and P38 MAPK dependent manner [91]. Accumulating evidence reveals the possible antioxidant induction ability of sulforaphane, an isothiocyante mostly rich in cruciferous vegetables, via the activation of Nrf2 signaling and protection from the devastating effect of oxidative stress. Molecular evidence revealed that sulforaphane significantly modulates Nrf2 signalling by activating the MAPK pathway and epigenetically altering the Nrf2 promoter [92]. The protective effect of different natural and synthetic flavonoids, such quercetin, fisetin, luteolin, eriodictyol, galangin, baicalein, EGCG, 3,6-dihydroxy flavonol, and 3,7 dihydroxy flavonol, against oxidative stress-mediated death in retinal pigment epithelial (RPE) cells was studied and it was observed that these compounds have the potential to stimulate Nrf2 expression and phase II detoxifying enzymes in RPE cells [93]. Baicalein has been shown to possess a radioprotective effect by activating the ERK/Nrf-2 signaling, thus mitigating radiation-induced hematopoietic injury [94].

### 3.4. JAK/STAT Pathway

The signal transducer and activator of transcription (STAT) and Janus kinase (JAK) pathways play an essential role in cytokine signaling and thus regulate multiple cellular responses, including cell survival and mortality, cellular differentiation, cell maintenance, hematopoiesis, and inflammatory responses [95]. The interaction of ligands, such as interleukin, growth factors, and hormones, through various transmembrane receptors regulates their activity and controls cellular response. Ligand binding causes dimerization of the receptors and leads to the auto/transphosphorylation at specific tyrosine residues of the c terminal tail of the receptor. The phosphorylated tyrosine residues serve as a docking site at its –SH domain-containing the STAT molecule. Once the STAT molecules are phosphorylated at specific conserved tyrosine residues, they act as transcription factors that dimerize and translocate to the nucleus. They bind specific promoters and modulate downstream gene expression involved in cellular proliferation, differentiation, and apoptosis [96,97]. IR modulates the JAK/STAT pathway and plays a major role in regulating the immune response associated with radiation toxicity. Studies have also shown that STAT proteins are essential in IR induced stress [98]. Previous in vitro and in vivo studies also revealed that IR modulates the expression of different cytokines, immune modulators, and growth factors, including IL-1α, TNF-α, IL-6, IL-1β, type I IFN, GM-CSF, IL-4, IL-5, IL-10, IL-12, IL-18, and TGF-β [99,100,101]. Previous reports demonstrated that the inhibition of the STAT3 increases the radiation sensitivity of tumor cells, thus mediating radiation-induced apoptosis in different cell lines [102,103]. An increase in the concentration of cytokines, including IL-1β, TNF-α, IL-8, IL-6, or TGF-β, plays a prime role in modulating IR mediated response and encourages inflammation, cancer cell invasion, and radiation-induced fibrosis [104]. Targeting different chemical compounds gives rise to the effective radiosensitivity of tumors cells without apparent toxicity, which implies that JAK/STAT signaling will be a prominent molecular target by different phytochemicals to boost apoptosis of tumor cells.

A study by Chung and Vadgama et al. revealed that the activity of curcumin and EGCG can suppress STAT3 and NF-kβ signaling at a concentration of 10 µM in breast cancer stem cells [105]. Subsequent findings by Blaskovich et al. in 2003 revealed the potent inhibitory activity of cucurbitacin I on phosphorylation of tyrosine resides in STAT 3 and JAK in human and mice cancer cell lines [106]. Later on, the study showed the prospective inhibitory effect of cucurbitacin on JAK/STAT signaling [107]. The impact of resveratrol in modulating the JAK-STAT pathway is well understood. It has been observed that the compound blocks or inhibits the phosphorylation of JAK and several STAT proteins in different cell lines, thus regulating the expression of several anti-apoptotic proteins [108,109]. Resveratrol can inhibit the src tyrosine kinase activity and block the JAK/STAT pathway of tumor cells [110]. Studies have also documented the therapeutic activity of curcumin in multiple myeloma cells, where the compound inhibits STAT3 phosphorylation and significantly prevents its translocation to the nucleus [111]. Curcumin also reduced the expression of cell proliferative genes, such as Bcl-XL, cyclin B1, and molecules involving cell invasion (VEGF, MMP2, MMP7, and ICAM), by inhibiting STAT3 phosphorylation [112]. The potential role of EGCG in suppressing the STAT3 phosphorylation has been elucidated and studies have shown that the compound significantly inhibits STAT3 phosphorylation and its activity in different cell lines [113,114]. The suppressing ability of caffeic acid and its derivative CADPE on the tumor angiogenesis was studied and the result indicates that each compound prevents VEGF expression by blocking STAT3 phosphorylation [115].

### 3.5. Agents Targeting P53 Signaling Pathway

P53, a prominent protein also known as “guardian of the genome”, plays a significant role in radiation signalling pathways. The protein is stimulated by different types of stressors, e.g., IR, hypoxia, carcinogenesis, and oxidative stress. Under physiological conditions, P53 concentration is low. However, it increases under the influence of IR. It is then translocated into the nucleus from the cytoplasm where it modulates several downstream signalling molecules and thus plays a significant role in the regulation of cell cycle, DNA repair, and apoptosis, which promotes cell survival and differentiation [116]. The DNA damage response (DDR) pathway is the most effective signalling network which is activated upon exposure to IR and studies have shown that DNA double-stranded breaks (DSB) are created mainly by IR. Under normal conditions, the binding of Mdm2 to P53 promotes its ubiquitylation, whereas DNA damage activated several protein kinases that phosphorylate P53, thus reducing its affinity for Mdm and decreases the degradation of p53. Activated P53 activates its downstream effector P21, which regulates G1/S-Cdk and S-Cdk complexes and maintains cell cycle arrest at the G1 phase [117]. Depending on the severity of damage in a coordinated and precise manner, P53 may activate DNA repair genes or induce the expression of Bcl2, Bax, and Caspase3, which promote apoptosis.

Interestingly, P53 mutations are significantly associated with the resistance of many cancer cells to several anticancer agents [118]. However, evidence shows that the expression of wild-type P53 in response to radiation stimuli, mutant P53 is constitutively produced in response to radiation [119]. DNA damage by IR as well as UV radiation has been reported to stimulate the P53 activity [120]. Different stress stimuli, e.g., ATM, ATR, and checkpoint kinases like Chk1 and Chk2, specially modulate the phosphorylation of several post-translational modifications, which control the interactions P53 and MDM2 and thus influence the stability of P53 [121]. The inhibitory activity of quercetin on P53 level was shown in Dalton’s lymphoma mice and it has been observed that quercetin modulates the PI3k-Akt-P53 signaling pathway via the downregulation of P53 and activation of Akt [122]. Furthermore, the apoptosis-inducing ability of resveratrol has been reported in MDA-MB-231 cancer cell lines and it significantly reduces the expression of PI3K/Akt while stimulating the expression of cleaved caspase-9, P53, and cleaved caspase-3 [123].

Furthermore, cyanidin reverses cisplatin-induced apoptosis ability in HK-2 proximal tubular cells by modulating P53 phosphorylation [124]. Naringin downregulates the activation of P53 and thus suppresses the cisplatin-induced nephrotoxicity in the rat models [125]. According to evidence, paeonol, a phenolic molecule derived from the bark of the *Moutan cortex*, the root bark of the Chinese peony tree, strongly suppresses the production of P53, acetyl H3K14, and H4K16, which are elevated by H_2_O_2_-mediated oxidative stress, [126]. A further study shows that indole-3-carbinol (I3C) present in cruciferous plants like broccoli significantly inhibits p53–MDM2 interaction, thus leading to apoptosis [127]. Interestingly, it was observed that inhibition of STAT3 activity by Penta-1,2,3,4,6-O-galloylbeta- D-glucose is P53 dependent in prostate cancer cells in vitro and defeats prostate xenograft tumor growth in vivo [128]. Earier studies revealed that γ irradiated fibroblast cells show resistance to caffeine, due to the ATM-dependent phosphorylation of p53 [129]. Interestingly, later on, it was observed that caffeine inhibits gamma and UV radiation-induced phosphorylation of Ser15 and p53 residues in the ATM signalling pathway [130]. A recent finding shows the potential affinity between NTD (N terminal domain) of p53 and EGCG, which could be implicated in targeting p53 during radiation response [131]. Figure 3 shows different phytochemicals and their radiation targets inducing different signaling pathways.

### 3.6. Notch Signaling

Notch signaling is an evolutionary conserved regulatory pathway that controls cell differentiation, proliferation, apoptosis, and other biological processes. Notch 1–4, which are single-pass transmembrane proteins, are found in mammals. Delta/Serrate/Lag-2 (DSL), Jagged 1 and 2, and Delta1, 3, and 4 are single-pass transmembrane proteins expressed on adjacent cells that interact with transmembrane ligand. The interaction between Notch receptors and membrane-bound ligands is an important prerequisite for signaling response. The binding of ligands leads to the cleavage and release of Notch intracellular domains (NICD), which subsequently translocate to the nucleus and interact with its promoter elements to modulate cellular responses [132]. Notch signaling has been associated with radiation resistance in glioma cells [133] and breast cancer cells [134]. Radiation resistance has also been observed in non-small lung cancer cells where the upregulation of Notch signaling plays a vital role [135]. In vivo and in vitro studies revealed that radiation decreases osteoblast differentiation and proliferation, mediating cell cycle arrest, impairs collagen synthesis, and induces apoptosis, thus impairing bone formation [136,137]. Thus, increasing Notch 1 and Notch 2 expression in osteoblasts will be ideal for preventing radiation-induced bone loss [138]. Genome-wide expression profiling indicates the inhibitory effect of genistein on Wnt 5a and Notch 2 expression in rat mammary epithelial cells and CML patients having altered tyrosine kinase activity [139]. Curcumin in combination with piperine inactivates Notch by decreasing Notch 1 expression, thus reducing mammosphere formation in cancer. The inhibitory activity of resveratrol on Notch has also been reported in lymphoblastic leukemia cells [140,141]. Curcumin also down-regulates Notch1 expression and its downstream signaling molecules in different types of cancer cells. Similar effects have been observed with diallyl trisulfide, which decreases Notch downstream genes [142]. Soy isoflavone upregulates the Notch1 and Hes5 in the cerebral cortex and prevents radiation-induced apoptosis [143]. Results from a clinical study have shown that only gamma-secretase inhibitors alone or in combination with chemotherapeutic agents are effective in inhibiting the notch signaling. However, an association of gastrointestinal toxicity and cardiotoxicity prompted further validation in clinical applications [144]. Table 1 shows the effect of different phytocompounds and their possible roles in radioprotection through different pathways.

### 3.7. Hedgehog Signaling

Hedgehog (Hh) signaling is an intricate signal transduction mechanism that plays a vital role in maintaining cellular proliferation, cell fate determination, embryonic development regulation, and tissue homeostasis. Deregulation and impairment of the pathway are implicated in stem cell renewal, congenital disability, and progression into various cancers. The seven-helix transmembrane (7TM) protein smoothened (SMO) plays a significant role in this process. In the absence of Hh ligand, SMO is inhibited by transmembrane protein patched (PTC). However, the binding of Hh to its receptor PTC facilitates the signaling of its downstream transcription factors such as Gli1(glioma-associated oncogene) and other transcription factors, thus regulating the expression of its target genes [145]. Studies on PTCH mutant mice indicate UV and IR-mediated basal cell carcinomas (BCC) by Hh target gene activation play a significant role in BCC tumorigenesis [146]. Studies have shown that mice with Ptch1+/− suffer from X-ray-induced cataracts more than the Ptch1+/+ mice, suggesting the upregulation of ptch1 in normal mice, which eradicates radiation-induced cataracts [147].

Further studies revealed the protective effect of sonic Hh signaling in human hepatocellular carcinoma cells against ionizing radiation [148]. Radiotherapy induced activation of Hh transcription factor Gl1 expression by the mTOR/S6K1 pathway in head and neck squamous cell cancer (HNSCC) and its blockade by cyclopamine suggest the role of Hh signaling in stromal resistance during radiotherapy [149]. Interestingly, a study shows the role of activated Hh signaling in tumor repopulation after radiotherapy [109]. Radiation stimulates the secretion of the Hh ligand, which can bind to the PTCH receptor, resulting in activation of GL1, thus regulating the progression of radiation-induced fibrosis in hepatic cells [150]. Baicalin, a flavonoid, activates the sonic Hh pathway by stimulating the expression of sonic Hh, SMO, and Gli1 proteins [151]. Genistein downregulates the Hh-Gli1 signaling pathway and reduces mammosphere formation [152]. In another study, long-term exposure to low doses of genistein showed the sensitization of breast cancer cell lines to radiation and decreased stem cell growth and mammosphere formation [153]. Cyclopamine targets Hh signaling by inhibiting SMO activation and could play a significant role in hindering various cancers [154]. The inhibitory role of curcumin on Hh-GlI signaling has also been reported [154]. Further studies in rat models showed the valuable effects of panaxotriol saponin in the upregulation of VEGF and angiopoietin-1 expression via the sonic hedgehog signaling pathway which protects against radiation-induced brain injury [155].

### 3.8. PI-Akt Signaling

The phosphatidylinositol-3-kinase (PI3K)/Akt pathway plays a significant role in controlling many pathological and physiological conditions governing various body processes, e.g., cell survival, cellular proliferation, angiogenesis, cellular metabolism, and differentiation. Inhibition of the PI3K pathway significantly decreases cellular survival, promoting apoptosis, whereas the activation of PI3K blocks apoptosis. Evidence indicates that activation of the PI3K/Akt signaling causes radiation resistance in cancer cells, whereas in normal cells, it shows a radioprotective effect [156]. Activation of Akt is associated with cell radioresistance and it has been observed that Akt causes the upregulation of several proteins, e.g., IKKα, mTOR (mammalian target of rapamycin), and Mdm2, in turn causing cell survival, growth, DNA repair, and cellular proliferation, as well as the downregulation of Bad, Apaf/Caspase 9, GSK3β, and p27, thus leading to downstream signaling events inducing cell cycle arrest and apoptosis [157]. Studies show that the activation of epidermal growth factor receptors in response to IR in multiple tumor types in vitro induces the activation of RAF-1-MEK1/2-ERK1/2 and the PI3Kphosphoinositide-dependent kinase-1-AKT pathways [158]. Moreover, studies have shown the association of the PI3K/AKT pathway with metabolic response during chemotherapy and RT [159]. Therefore, it is crucial to identify putative targets that cross-talk with other signaling cascades to achieve an efficient response in radiation-induced normal and carcinogenic cells. Several phytochemicals have been reported to play a tremendous role in regulating PI3K/Akt signaling, thus controlling cell survival during radiation stress by reducing apoptosis. Benzyl isothiocyanate (BITC), found in *Alliaria petiolata*, *Salvadora persica,* and other plants, has been found to prevent tumor growth by inhibiting the PI3K/AKT/FOXO pathway [160]. Soy isoflavones, such as genistein and daidzein, were shown to inhibit the IGF-1R/p-Akt, NF-κB, APE1/Ref-1, and HIF-1α signaling intermediates and sanitize the tumor cells to RT [161].

**Table 1 antioxidants-11-00049-t001:** Effect of phytocompounds and their possible role in radioprotection via different Signaling pathways.

Compound Name	Signaling Target	Effect/Possible Role	Reference
Allicin	JNK pathway	Downregulate ICAM-1 expression	[162]
Apigenin	Nf-kβ pathway	Modulate p53, p21, Bax caspase3 & 9	[163]
Arctiin	Wnt, MAPK pathway		[164]
Baicalein	Nrf2 pathway	Stimulates ERK & Nrf2 activity	[94]
Betullinic acid	Nf-kβ pathway	Act as a radio sensitizer in cancer cell	[27]
Caffeine	p53 signaling	Increases ATM activity	[129]
Carvacrol	TNF α signaling	Decreases radiation induced oxidative stress	[165]
Chlorophyllin	Nrf 2 & Nf-kβ pathway	Possesses antioxidant, antiapoptotic activity	[166]
Curcumin	Notch pathway Nrf2 pathway	Decreases Notch 1 & 2 activity Induces PI3K, ERK, HO-1, P38-MAPK	[140,167] [83,168]
DIM	Nf-kβ pathway	ATM, DBR	[152]
Diosmin	Wnt/β-catenin pathway	Increases PPARγ expression & possess antioxidant, anti-inflammatory, anti-apoptotic property	[67]
Diospyrin (Diospyrin dimethylether)	P53 and Nf-kβ pathway	Downregulate COX-2, Bcl-2, Upregulates p53, p21	[169]
EGCG	Nrf2 pathway	Induces PI3K, ERK, HO-1, P38-MAPK	[83,168]
Ferulic acid	c-JNK, ICAM-1, VCAM-1 mediated signaling	Antioxidant and Anti-inflammatory Activity	[170]
Fucoidan	TGF-β, Smad pathway	Inhibits TGF-β, Smad activity	[171]
Genistein	Hedghog pathway Notch pathway	Down regulate Hedgehog-GLI 1 Activity Decreases Notch 1 & 2 activity	[152] [140,167]
Hesperidin	Nf-kβ pathway	Increases COX2 & NO activity	[172]
Lycopene	Nf-kβ, JAK-STAT pathway	Possesses antioxidant, anti-inflammatory activity Inhibits NF-kB, p65, STAT3, IL-6, TNF-α, COX2, PGE2	[173]
Mangiferin	Nrf2 pathway	Increases NOQ1 level	[174]
Melatonin	Nf-kβ, PI-Akt pathway	Decreases p-AKT, p-ERK, COX2, p65	[175]
Parthenolide	Nf-kβ pathway	Inhibit NF-KB signaling	[176]
Piperine	Notch pathway	Decreases Notch 1 & 2 activity	[140,167]
Quercitin	Nf-kβ pathway	Inhibits ERK and p38	[177]
Resveratol	Nf-kβ pathway Notch pathway	Decreases NF KB signalling of p65 & IKB kinase activity	[55] [140,167]
Rutin	PI3K/AKT/GSK-3β/NRF-2-pathway	Increases p-PI3K, p-AKT and p-GSK-3β activity	[178]
Saponin	Hedgehog pathway	Up regulate VEGF & Angiopoetin1	[178]
Soya isoflavon	Notch pathway	Up regulate Notch 1 & HES 5 activities	[179]
Sulphora phane	Wnt/β-catenin pathway	Down regulate Wnt/B Catenine activity	[74]
Thymol	TNF α signaling	Decreases radiation induced oxidative stress	[165]
Ursolic acid	Nf-kB and JNK pathway	Decreases Nf-kB, IL-1β, TNF-α, IL-6	[179]
Vanillin	P53-NOXA pathway	Decreases p53 activity	[180]
WGP	Nf-kβ, P38-MAPK pathway	Decreases level of ROS & RNS Production	[161]

## 4. Conclusions

The last decade of research has focused on exploiting molecular mechanisms involved in the bystander response using novel pharmacologically active agents of clinical importance to ameliorate adverse effects caused by radiotherapy. Rapid technological innovation, natural/artificial radiation exposure, nuclear accidents, treatment, and medical imaging, among others, are all sources of radiation exposure. These planned and/or unintentional exposures to radiation may lead toxicities and severe radiation-related diseases. Protecting the biological system from the detrimental effects of radiation is critical. Decades of research have resulted in the development of various synthetic and semi-synthetic chemicals to ameliorate the dangers of radiation. However, because of their toxicity and side effects, alternate sources are critical. Previous studies have revealed that the Indian system of traditional medicine offers a wide range of pharmacologically active substances, including anti-inflammatory, anti-mutagenic, antioxidant, free radical scavenging, and radioprotective properties. We have highlighted the potential radioprotective properties of some phytochemicals that exert their actions by modulating the different signaling pathways. However, there is a translational gap in the use of this therapeutic arsenal of chemicals from bench to bedside. Furthermore, the inability of the compounds to distinguish between normal and malignant cells renders them therapeutically unsuitable. Future research should focus on adapting the in-silico method, utilizing high-throughput technology, developing an acceptable study design to conduct desirable clinical trials, chemical surface modification drug repurposing (drug repositioning), and generating a noncomplex structure. Not only will the techniques mentioned above reduce toxicity, but they will allow to distinguish between normal and malignant cells, allowing for improved therapeutic options.

## Figures and Tables

**Figure 1 antioxidants-11-00049-f001:**
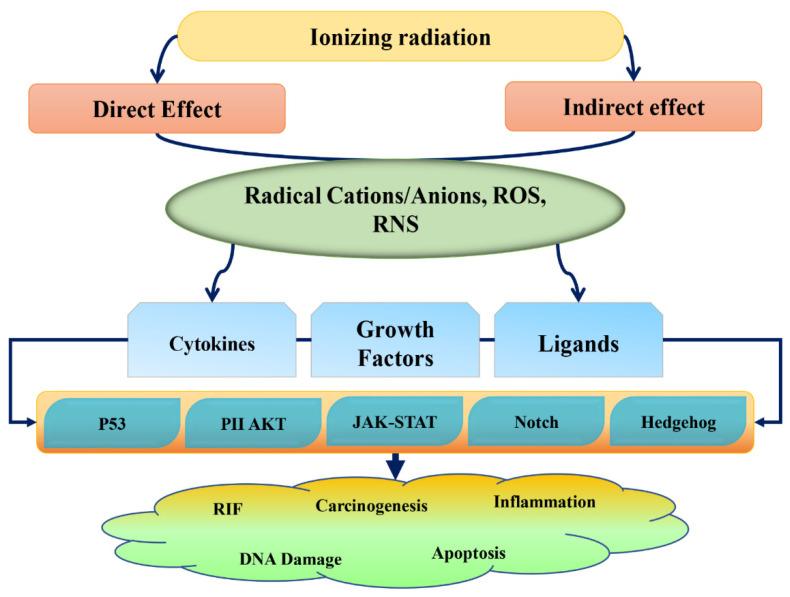
Effect of radiation on the biological systems.

**Figure 2 antioxidants-11-00049-f002:**
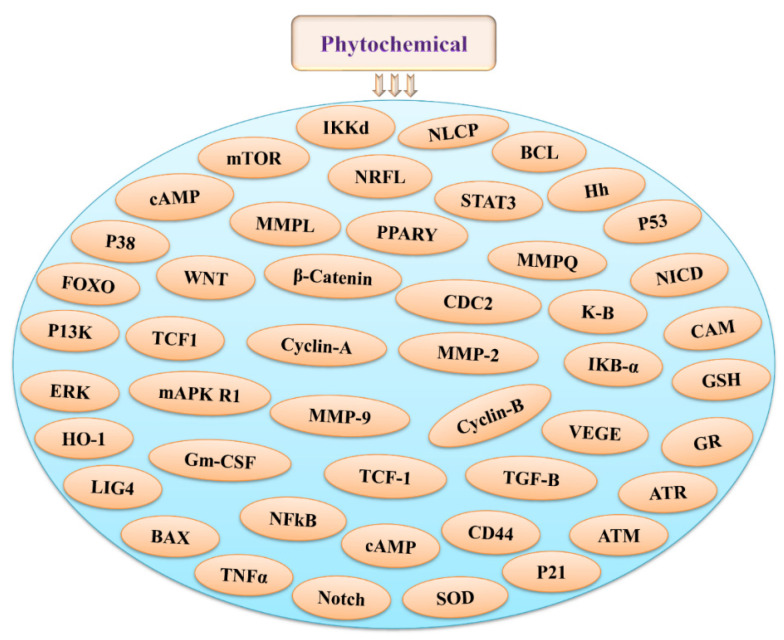
Different intermediates as a potential targets for phytochemicals during radiation-induced stress.

**Figure 3 antioxidants-11-00049-f003:**
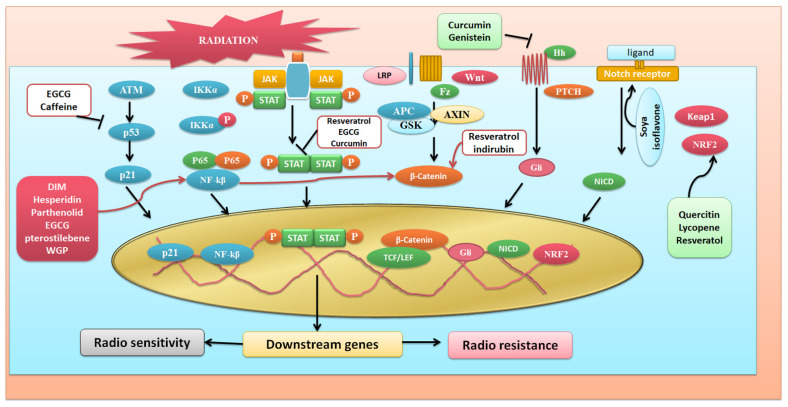
Phytochemicals and their targets on different radiation induced different signaling pathways.

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
