# Peer review of "Phytochemicals: Potential Therapeutic Modulators of Radiation Induced Signaling Pathways"

_antioxidants, 2021, doi:10.3390/antiox11010049_

Round 1
Reviewer 1 Report
This review is of scientific interest, as it provides information on naturally-occurring chemical agents that can help alleviate the deleterious effect of ionizing radiation through its antioxidant effect. However, English must be revised for a more fluent and understandable reading. In addition, in different places in the manuscript the speech is blocked or interrupted by sentences out of context, which make communication not fluid. Also, there are typographical errors that also affect this. For this reason, the entire text should be reviewed to improve the expression in English and eliminate or correct those elements that hinder the common thread of the discourse. Likewise, there is some concept that has to be redirected because it is not exactly correct.
See the comments and recommendations for improvement in the attached file.

Reviewer 2 Report
The protection of cells and tissues from ionizing radiation damages, as well as the radiosensitization of the cancer cells or the inhibition of cancer growth, are all topics of paramount importance. The application of phytochemicals to the solution of these problems is a research field developed for a long time worldwide and it obtained some important results; The information presented in your review paper can stimulate this research. Nevertheless, your manuscript has two main drawbacks:
- it must be deeply revised concerning the English language use since several sentences are very difficult to understand at first reading.
- the structure of your text is, of course, correct from a certain point of view, but it appears confusing to a reader looking for the properties of the different phytochemicals. Your paper clearly highlights the cellular pathways which can be affected by ionizing radiation and are involved in cellular homeostasis, but in my opinion, it is not helping to list together the effects of the different compounds according to these pathways and I suggest you order the information in a different way, preparing paragraphs each one concerning a specific natural compound and listing there all its antioxidant, radioprotective, radiosensitizing, anticancer properties. Moreover, I suggest you to underline in a particular way the results obtained in clinical studies, if any.
Round 2
Reviewer 1 Report
The authors have made the pertinent modifications concerning the major changes and some of the minor changes requested. However, it is still necessary to attend to the consideration of many suggestions for improvement and completely recheck all the text, which allow a higher quality of the manuscript to be achieved.
The document is of scientific interest, but it must improve the expression and correct typographical and fatal grammar mistakes. In this review, again multiple proposals for improvement are suggested for a better understanding of the text. These proposals have been made up to line 254. As commented in the previous review, it is in the authors' hands to review the rest of the document carefully (between lines 255-519), in order to correct expression deficiencies that may allow the document to be really understandable. Please, include commas in the appropriate places, add or use appropriately the verbs, use appropriate binding links to conveniently connect sentences, check the singular and plural, etc. Finally, the authors should look at the beginning of the conclusions, where it seems that line 497 is unnecessary.

Reviewer 2 Report
The revised version of the manuscript was significantly improved concerning the English language use, which helps in understanding the text. By reading carefully this new version you can still find various sentences needing correction (for example by splitting ) to make them easier to read.
I agree with the Author when it mentions the several reviews on the different properties of phytochemicals and with his choice to underline their effects on signaling pathways, but he must revise each paragraph reducing it to essential, avoiding repetitions, and more clearly highlighting that their radioprotective or radiosensitizing effects have no therapeutic usefulness during cancer radiotherapy, perhaps in protection from irradiation-induced disease, if any. You state it just in the Conclusion, but it would be more useful in the Introduction.
Few more particulars:
line 73: cancer prevention or treatment?
Line 80: H and H3O+ are not considered radicals, usually, as well as eaq is not a ROS, even obtained from water radiolysis.
I suppose that the title of paragraph 3.3 must be different from that of 3.2.
